# TAP-CT: 3D Task-Agnostic Pretraining of Computed Tomography Foundation Models

**Tim Veenboer[1] George Yiasemis[1,2] Eric Marcus[1,3,*] Vivien Van Veldhuizen[1] Cees G. M. Snoek[2] Jonas Teuwen[1] Kevin B. W. Groot Lipman[1]**

{T.VEENBOER, K.GROOT.LIPMAN}@NKI.NL

[1]*Netherlands Cancer Institute,* [2]*University of Amsterdam,* [3]*Kaiko,* [*]*Contributed while at NKI*

## Abstract

Existing foundation models (FMs) in the medical domain often require extensive fine-tuning or rely on training resource-intensive decoders, while many existing encoders are pretrained with objectives biased toward specific tasks. This illustrates a need for a strong, task-agnostic foundation model that requires minimal fine-tuning beyond feature extraction. In this work, we introduce a suite of task-agnostic pretraining of CT foundation models (TAP-CT): a simple yet effective adaptation of Vision Transformers (ViTs) and DINOv2 for volumetric data, enabling scalable self-supervised pretraining directly on 3D CT volumes. Our approach incorporates targeted modifications to patch embeddings, positional encodings, and volumetric augmentations, making the architecture depth-aware while preserving the simplicity of the underlying architectures. We show that large-scale 3D pretraining on an extensive in-house CT dataset (105K volumes) yields stable, robust frozen representations that generalize strongly across downstream tasks. To promote transparency and reproducibility, and to establish a powerful, low-resource baseline for future research in medical imaging, we will release all pretrained models, experimental configurations, and downstream benchmark code at https://huggingface.co/fomofo/tap-ct-b-3d.

**Keywords:** CT, Foundation Models, Self-Supervised Learning, 3D DINOv2

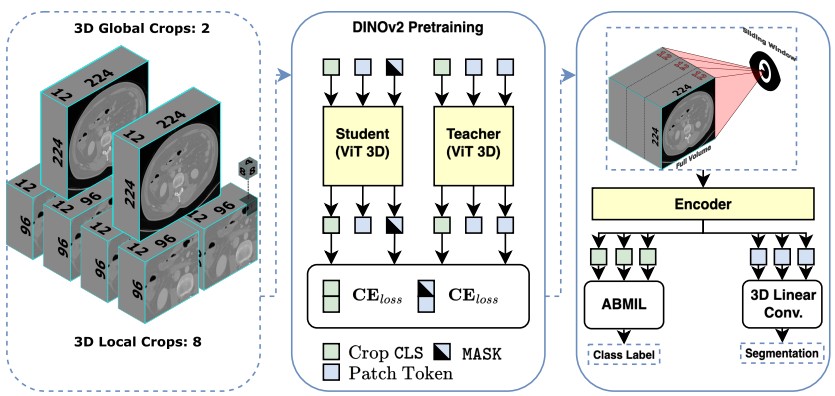

Figure 1: **Pretraining and Evaluation of TAP-CT Foundation Models.** Models are pretrained using a novel 3D adaptation of the DINOv2 framework and subsequently evaluated based solely on the representational quality of their learned features.

## 1. Introduction

Computer vision in medical imaging faces significant challenges hindering widespread adoption of AI in healthcare, particularly the lack of annotated data (Zhou et al., 2021b). At the same time, hospitals typically maintain substantial repositories of imaging data. Computed Tomography (CT) is among the most widely used imaging modalities, especially in cancer care. CT volumes are generally highly homogeneous: each scan consistently provides a partial or full 3D view of a patient's body. Variability across CT scans is relatively limited, as most anatomical structures appear consistently present across individuals. In principle, an AI model should be able to learn latent representations of these structures while simultaneously capturing subtle patient-specific abnormalities within the same representational space. These characteristics make CT an ideal candidate for self-supervised learning (SSL).

Yet, current applications of SSL in CT imaging are largely limited to either (1) approaches that require fine-tuning of the encoder or an extensive decoder to extract useful information from the learned representations (Pai et al., 2025; Wu et al., 2024), or (2) methods that use the encoder only for specific tasks (Pai et al., 2024; He et al., 2025; Li et al., 2024b), as the pretraining objective biases the model toward those tasks. Moreover, existing approaches are often pretrained on the same datasets used for evaluation, due to limited data availability. The objective of this work is to establish a CT-native foundation model whose representations remain robust across downstream tasks, providing a reliable baseline that avoids task-specific pretraining biases and eliminates the need for heavy fine-tuning.

DINOv2 aligns global representations using the DINO objective (Caron et al., 2021), attracting embeddings from 2D local and global views, while learning local representations through masked image modeling in representation space, inspired by iBOT (Zhou et al., 2021a). We reinterpret DINOv2's local and global crops for volumetric data by introducing a GPU-based 3D random resized crop. We also redesign the masking strategy to be compatible with 3D inputs. In addition, we adapt an existing Vision Transformer (ViT) (Dosovitskiy et al., 2020) implementation to handle volumetric data by modifying the patch embedding layer and the positional encoding grid. Using this implementation, we train a family of 2D and 3D ViTs with DINOv2-style pretraining on a large-scale, in-house dataset of 105K CT volumes to obtain high-quality feature extractors. Across a comprehensive set of public segmentation and classification benchmarks, we show that our models achieve state-of-the-art performance in frozen-feature segmentation using a linear decoder, while classification results highlight limitations in full-volume representation learning. The key contributions of this work are summarized below:

- We introduce *TAP-CT*, a family of 6 task-agnostic foundation models for CT pretrained on a large in-house dataset of 105K CT volumes. We publish the pretrained weights to provide a strong, low-resource baseline for the medical imaging research community.

- We adapt the DINOv2 SSL framework and ViT architecture to handle volumetric inputs, facilitating comprehensive 3D pretraining on CT data.

- We release the code and configurations for benchmarking our FMs and other existing approaches to promote simple and standardized evaluation of vision foundation models in medical imaging.

## 2. Related Work

**Transfer learning:** Prior work in transfer learning has demonstrated the effectiveness of transferable representations in CT imaging for both segmentation (Gao, 2024; Karimi et al., 2021) and classification (Kim et al., 2022). SuPReM (Li et al., 2024b) further shows that transfer learning can substantially improve data efficiency and model performance in low-label regimes for CT segmentation. However, the models are pretrained with a supervised segmentation objective, resulting in a feature space that is heavily biased toward this task.

**Biomarkers:** FMCiB (Pai et al., 2024) extracts 50 mm$^3$ patches centered on a large set of lesions and applies SimCLR (Chen et al., 2020) to contrast lesion patches against non-lesion patches. This approach effectively models imaging biomarkers and achieves strong performance on downstream tasks such as nodule malignancy classification. Despite this, FMCiB focuses on small patches and is therefore unlikely to capture generalizable global representations.

**Promptable image segmentation:** Numerous adaptations of SAM (Kirillov et al., 2023) have been proposed for CT imaging (Ma et al., 2024; Zhang and Liu, 2023; Cheng et al., 2023, 2024). VISTA3D (He et al., 2025) aggregates 2D SAM embeddings from multiple CT views into a 3D supervoxel representation, on top of which an encoder is subsequently trained. However, similar to SuPReM, its representations are still shaped by segmentation tasks.

**General CT foundation models:** VoCo (Wu et al., 2024) leverages geometric patterns inherent in CTs within a contrastive learning paradigm to derive latent representations via a SwinUNETR encoder (Hatamizadeh et al., 2022). The authors establish that VoCo surpasses current supervised approaches through pretraining on large-scale CT datasets. However, this performance is achieved by fine-tuning both the encoder and a task-specific SwinUNETR decoder. CT-FM (Pai et al., 2025) uses a SegResNet (Myronenko, 2019) encoder pretrained with SimCLR on large public datasets. While the model demonstrates strong overall performance, it relies on end-to-end fine-tuning with a SegResNet decoder for segmentation.

**DINOv2 in medical imaging:** Several studies have investigated the capabilities of the original DINOv2 ViTs on medical imaging tasks (Hussien et al., 2025; Baharoon et al., 2024). The DINOv2 framework has also been applied to x-ray imaging (Pérez-García et al., 2025). X-ray imaging which is inherently 2D, and thus requires no adaptations of the original framework. More recently, Curia (Dancette et al., 2025) employed regular DINOv2 training on a large-scale dataset of CT and MRI slices. In contrast, our work translates DINOv2 to the 3D domain and demonstrates the benefits of 3D over 2D pretraining, as reported in prior non-FM studies (Avesta et al., 2023; Çiçek et al., 2016).

Collectively, prior work highlights both the potential and the fragmentation of CT foundation models: many approaches require extensive end-to-end fine-tuning, are task-specific, or restrict pretraining to 2D slices. On the other hand, our goal is to develop a large-scale, task-agnostic, 3D-pretrained CT foundation model that produces off-the-shelf, readily usable features.

## 3. Methodology

We first describe our adaptations to DINOv2 and ViTs from 2D to 3D, followed by the dataset and preprocessing. Finally, we present an overview of the ten downstream tasks.

Table 1: Downstream task details for each publicly available model and ours. For Curia, volumes are resized to $(z, 512, 512)$. For TAP-CT we evaluate both on $(z, 224, 224)$ and $(z, 512, 512)$. An asterisk (*) indicates that only the encoder is used.

| Model | Architecture | Params | Spacing | Window Size | Data Size |
|---|---|---|---|---|---|
| **CT-FM** | SegResNet* | 77.8M | (3.0, 1.0, 1.0) | (24, 128, 128) | 148K |
| **Curia** | ViT Base (2D) | 86.0M | - | (1, 512, 512) | 150K |
| **FMCiB** | ResNet | 184.5M | (1.0, 1.0, 1.0) | (64, 64, 64) | 11.5K |
| **SuPReM** | UNet* | 19.1M | (1.5, 1.5, 1.5) | (96, 96, 96) | 2.1K |
| **VISTA3D** | SegResNet* | 175.0M | (1.5, 1.5, 1.5) | (128, 128, 128) | 11.5K |
| **VoCo** | SwinUNETR Base* | 53.2M | (1.5, 1.5, 1.5) | (96, 96, 96) | 160K |
| **TAP-B-3D (ours)** | ViT Base (3D) | 86.0M | - | (12, 224, 224) | 105K |

### 3.1. Pretraining setup

**DINOv2:** DINOv2 is a joint-embedding architecture in which a student model learns to construct informative latent representations by comparing its outputs with the teacher model, which is a momentum-updated replica of the student. The framework uses two objectives: the DINO (Caron et al., 2021) loss aligns representations of local and global crops for robust image-level representation, while the iBOT (Zhou et al., 2021a) loss leverages a subset of `[MASK]` tokens in the student's global crops. Here, the teacher processes all tokens, and the student has to match the teacher's representations for the masked tokens. We refer to (Oquab et al., 2024) for comprehensive details.

**Local and global crops:** Random resized cropping enables creation of local and global crops. In 2D, this transformation stochastically selects a crop based on area, sampling random aspect ratios, and resizing the crop to a fixed target size. In 3D, implementing random resized cropping requires a volumetric region, which faces two constraints: computational limitations caused by interpolation of the crop, and the varying number of slices per CT. The axial dimension is often $(512, 512)$, but $z$ is inconsistent relative to height-width thus sampling a height-depth ratio becomes non-trivial.

A more practical and effective solution, adopted in this work, selects an area and its aspect ratio in the axial plane and extends the crop by a fixed number of slices along the depth axis. This approach also acts as an implicit augmentation, since the physical slice spacing in world-coordinate $z$-space varies across CT volumes. To mitigate the substantial overhead of CPU-bound interpolation, we implement the 3D random resize crop on GPU.

**Masking strategy:** We also adapt the masking strategy of DINOv2 to volumetric data. For each global crop, we randomly sample multiple masked regions until the desired number of masked patches is reached. In 2D, an area and subsequently a height–width aspect ratio are sampled for each region. In contrast to our crop sampling, we extend masking to 3D by sampling a height–depth aspect ratio, since depth does not vary across crops. Consequently, the masks tend to be more cube-like, with height–width and height–depth aspect ratios that are relatively similar.

**Pretraining hyperparameters:** The training regimen mostly follows (Oquab et al., 2024). We train for 125,000 iterations with batch size 2048 on 8 H100 GPUs. We observed that increasing the learning rate warmup phase from 12,500 to 25,000 iterations was

critical to prevent early representational collapse. The full set of hyperparameters is listed in Appendix A.

**ViT adjustments:** We adapt the ViT by extending the patch embedding layer from a 2D to a 3D convolution. In addition, the learned positional encoding is interpolated onto a 3D grid to account for positional variations along the z-axis.

**TAP-CT:** As a baseline, we train 2D models with a global crop size of (224, 224), a local crop size of (96, 96), and a patch size of (16, 16), following (Oquab et al., 2024). For 3D volume experiments, we consider two distinct configurations. The first configuration employs a global crop size of (6, 224, 224), a local crop size of (6, 96, 96), and a patch size of (1, 16, 16), termed 2.5D, given its 3D volume and 2D patch size. The second configuration utilizes a global crop size of (12, 224, 224), a local crop size of (12, 96, 96), and a patch size of (4, 8, 8), termed 3D. The decision to keep $z$ consistent between global and local crops was derived empirically; see Appendix D. For each configuration, we train both ViT-S and ViT-B models. For brevity, we denote these models as TAP-S/B-2D, TAP-S/B-2.5D, and TAP-S/B-3D, respectively.

## 3.2. Dataset

The pretraining dataset comprises 104,405 CT volumes from 19,995 oncological patients, with a mean age of 63 years. The scans contain an average of 316 slices, totaling 32,973,620 slices across the dataset. The median voxel spacing is 0.79 mm × 0.79 mm × 1.5 mm along the x, y, and z axes, respectively. Distributions of patient age, sex, and scanner manufacturer are provided in Appendix B.

**Preprocessing:** We extract the mean, standard deviation, 0.5th and 99.5th percentiles of the foreground voxels in the dataset, following `nnUNetv2` (Isensee et al., 2021). We use these to clip and normalize the volumes. Exact values listed in Appendix B.

**SSL Augmentations:** Since CTs are single-channel, color-based augmentations used in DINO are replaced with a random gamma adjustment, while the random Gaussian blur transformation is retained. Prior research showed that for shorter training regimens, such as ours, data augmentations can potentially play a significant role in model performance (Moutakanni et al., 2024) which is why the augmentations are replaced rather than omitted. All augmentations are performed on the GPU to speed up training time.

## 3.3. Downstream tasks

To assess the performance of pretrained FMs (Table 1), we conduct a series of downstream segmentation and classification experiments. For each task, embeddings are first extracted using a sliding window approach (Cardoso et al., 2022), after which a single layer is trained to exclusively evaluate the representational strength of the encoder. All evaluations are implemented in EVA (kaiko.ai et al., 2024). The evaluation procedure is illustrated in Figure 1, while further details of downstream task specifics are provided in Appendix C.

### 3.3.1. SEGMENTATION

We leverage a single linear convolution layer that maps the frozen encoder embeddings to segmentation logits. Performance is measured using the macro-averaged Dice Similarity Coefficient (DSC) over all non-background classes. The datasets employed in this work are:

**TotalSegmentator v2** (Wasserthal et al., 2023), a full-body segmentation dataset annotated with 117 distinct anatomical structures. To simplify evaluation and reduce redun-

Table 2: Segmentation performance after fine-tuning a single linear convolution layer on frozen encoder embeddings. We report the mean Dice Similarity Coefficient (DSC) ± standard deviation averaged over three runs. **Best** results are bolded, while second-best results are underlined. OOM = Layer ran out of memory on an H100 80GB GPU. (†) = Model's pretraining data included these public datasets. (*) = Evaluated on ($z$, 224, 224): H100 80GB went OOM on ($z$, 512, 512).

| Model | AMOS22 (DSC) | LiTS17 (DSC) | KiTS23 (DSC) | TotalSeg. (DSC) | MSD Pancreas (DSC) | Average |
|---|---|---|---|---|---|---|
| Curia | 0.669 (± .006) | 0.571 (± .003) | 0.429 (± .006) | 0.425* (± .002) | 0.350 (± .003) | 0.489 |
| SuPReM† | 0.450 (± .004) | 0.440 (± .002) | 0.363 (± .002) | 0.353 (± .002) | 0.301 (± .000) | 0.381 |
| CT-FM† | 0.417 (± .008) | 0.416 (± .013) | 0.265 (± .006) | 0.317 (± .000) | 0.213 (± .009) | 0.326 |
| VISTA3D† | 0.364 (± .001) | 0.377 (± .014) | 0.243 (± .010) | 0.226 (± .000) | 0.161 (± .002) | 0.274 |
| VoCo† | 0.120 (± .004) | 0.345 (± .014) | 0.176 (± .004) | 0.072 (± .000) | 0.120 (± .001) | 0.167 |
| FMCiB | 0.061 (± .008) | 0.362 (± .002) | 0.110 (± .018) | OOM | 0.051 (± .002) | 0.146 |
| TAP-B-3D | **0.724** (± .001) | **0.626** (± .004) | **0.480** (± .005) | **0.651*** (± .001) | **0.429** (± .003) | **0.582** |

dancy, we merge certain classes, resulting in 49 distinct classes (Appendix C). **AMOS22** (Ji et al., 2022) has abdominal annotations for 15 different organs. **LiTS17** (Bilic et al., 2022) contains CTs for liver and liver tumor segmentation. **KiTS23** (Heller et al., 2023) is focused on kidney, kidney tumor and kidney cyst segmentation. **MSD Pancreas Tumor** (Simpson et al., 2019) contains labels for the pancreas and any associated lesions.

### 3.3.2. Classification

We evaluate classifications tasks using an Attention-Based Multiple Instance Learning (AB-MIL) head (Ilse et al., 2018). The ABMIL head is applied to the [CLS] embeddings when available, and to patch embeddings otherwise. The classification datasets are:

**LUNA16** (Setio et al., 2017; Armato III et al., 2015), a lung nodule malignancy task based on radiologists' verdict. For this task, each encoder extracts features from a 50mm³ crop centered around the lesion. The metric reported for this downstream task is area under the ROC curve (AUC). Given that a commonly used split (Pai et al., 2024) is on nodule level, and thus some patients/scans are both in training and test set, we use a new patient-level split. **LUNA25** (Peeters et al., 2025b,a) is another lung nodule malignancy task but with pathologically confirmed labels. A 50mm³ crop is extracted around each nodule. The reported metric is AUC. **RSNA2023** (Hermans et al., 2025) is a dataset focused on multi-label scan-level abdominal trauma classification. We consider only injuries to the kidney, spleen, and liver, and omit cases of extravasation and injuries to the bowel. The evaluation metric for this task is the micro-averaged multi-label AUC. **RSNA2022** (Lin et al., 2023) is a multi-label scan-level classification task to identify cervical spine fractures across vertebrae. Due to class imbalance, performance is evaluated using the micro-averaged multi-label Average Precision (AP). **FDG-PET-CT** (Gatidis and Kuestner, 2022) is a full-body dataset of patients diagnosed with three distinct cancer subtypes. For this downstream task, we used the diagnostic CTs and formulated a scan-level binary task, classifying tumor presence. The evaluation metric for this task is AUC.

## 4. Results & Analysis

The following section outlines the results obtained across downstream tasks, along with an analysis and ablation studies conducted in this work.

Table 3: Segmentation performance after fine-tuning a single linear convolution layer on frozen encoder embeddings, averaged over three runs. For each task, we report the mean Dice Similarity Coefficient (DSC) $\pm$ standard deviation. **Best** results are bolded, while second-best results are underlined.

| Model | Patch Size | Image Size | AMOS22 (DSC) | LiTS17 (DSC) | KiTS23 (DSC) | TotalSeg. (DSC) | MSD Pancreas (DSC) |
|---|---|---|---|---|---|---|---|
| TAP-S-2D | (16, 16) | (224, 224) | 0.545 ($\pm$ .002) | 0.513 ($\pm$ .002) | 0.387 ($\pm$ .002) | 0.482 ($\pm$ .001) | 0.301 ($\pm$ .004) |
| TAP-B-2D | (16, 16) | (224, 224) | 0.562 ($\pm$ .002) | 0.537 ($\pm$ .001) | 0.406 ($\pm$ .002) | 0.520 ($\pm$ .001) | 0.308 ($\pm$ .003) |
| TAP-S-2.5D | (1, 16, 16) | (6, 224, 224) | 0.553 ($\pm$ .001) | 0.537 ($\pm$ .001) | 0.435 ($\pm$ .004) | 0.508 ($\pm$ .003) | 0.315 ($\pm$ .001) |
| TAP-B-2.5D | (1, 16, 16) | (6, 224, 224) | 0.577 ($\pm$ .004) | 0.554 ($\pm$ .006) | 0.457 ($\pm$ .004) | 0.540 ($\pm$ .005) | 0.341 ($\pm$ .001) |
| TAP-S-3D | (4, 8, 8) | (12, 224, 224) | 0.633 ($\pm$ .003) | 0.572 ($\pm$ .001) | 0.445 ($\pm$ .003) | 0.612 ($\pm$ .001) | 0.373 ($\pm$ .003) |
| TAP-B-3D | (4, 8, 8) | (12, 224, 224) | **0.648** ($\pm$ .001) | **0.583** ($\pm$ .006) | **0.453** ($\pm$ .005) | **0.651** ($\pm$ .001) | **0.395** ($\pm$ .002) |
| TAP-S-2D | (16, 16) | (224, 224) | 0.700 ($\pm$ .001) | 0.551 ($\pm$ .002) | 0.447 ($\pm$ .002) | - | 0.390 ($\pm$ .003) |
| TAP-B-2D | (16, 16) | (224, 224) | 0.722 ($\pm$ .000) | 0.583 ($\pm$ .003) | 0.479 ($\pm$ .004) | - | 0.420 ($\pm$ .002) |
| TAP-S-2.5D | (1, 16, 16) | (6, 224, 224) | 0.699 ($\pm$ .002) | 0.574 ($\pm$ .004) | 0.496 ($\pm$ .006) | - | 0.412 ($\pm$ .003) |
| TAP-B-2.5D | (1, 16, 16) | (6, 224, 224) | **0.736** ($\pm$ .002) | 0.597 ($\pm$ .002) | **0.536** ($\pm$ .004) | - | **0.449** ($\pm$ .005) |
| TAP-S-3D | (4, 8, 8) | (12, 224, 224) | 0.711 ($\pm$ .001) | 0.584 ($\pm$ .008) | 0.458 ($\pm$ .005) | - | 0.422 ($\pm$ .003) |
| TAP-B-3D | (4, 8, 8) | (12, 224, 224) | 0.724 ($\pm$ .001) | **0.626** ($\pm$ .004) | 0.480 ($\pm$ .005) | - | 0.429 ($\pm$ .003) |

(The first six rows correspond to evaluation on $(z, 224, 224)$; the last six rows correspond to evaluation on $(z, 512, 512)$.)

### 4.1. Segmentation

The segmentation results of TAP-B-3D compared with publicly available pretrained models are presented in Table 2. TAP-B-3D shows strong downstream capabilities, achieving improvements of 5 to 23 percentage points in DSC over the next best model Curia. Table 3 summarizes the results for all TAP-CT models. TAP-B-3D, the largest model with the largest input context, performs best on (z, 224, 224) evaluation, while TAP-B-2.5D leads on 512-evaluation. Overall, segmentation quality largely scales with model dimensionality, model size, and input resolution.

### 4.2. Analysis

**Public foundation models**: Among the publicly available pretrained models, Curia achieves the strongest overall performance, which is expected given its use of task-agnostic pretraining through DINOv2. SuPREM attains the second-best performance, likely benefiting from its supervised pretraining with a segmentation objective on Abdomen Atlas 1.1 (Li et al., 2024a), which encompasses all downstream segmentation datasets considered in this work. The weaker performance of VoCo may stem from its pretraining design, which employs a large U-shaped decoder that relies on multilevel feature representations; as a result, its pretraining objective does not necessarily encourage the final-layer features to encode the bulk of the semantic information. FMCiB, which is pretrained on limited regions centered on lesions to extract specific biomarkers, is expected to underperform on full-volume segmentation tasks.

**The scaling of TAP-CT**: The substantial performance gap between TAP-B-3D, Curia, and other publicly available models highlights the need for general, high-capacity vision encoders in CT imaging that produce robust, standalone feature representations. The comparison in Table 3 of TAP-CT variants on volumes resized to $(z, 224, 224)$ – their native axial input resolution – demonstrates that foundation models for CT seem to adhere to conventional scaling laws: larger models consistently outperform smaller ones, and increased input context leads to improved results. The gains observed when moving from 2D to 3D further emphasize the importance of volumetric encoders over purely slice-based approaches.

Table 4: Classification results after fine-tuning an Attention-Based Multiple Instance Learning (ABMIL) model on frozen `[CLS]` embeddings, when available, or on frozen patch embeddings otherwise. Each metric (AP, AUC) is reported as mean ± standard deviation averaged over three runs. **Best** results are bolded, while second-best results are underlined.

| Model | LUNA16 (AUC) | LUNA25 (AUC) | RSNA2022 (AP) | RSNA2023 (AUC) | FDGPETCT (AUC) |
|---|---|---|---|---|---|
| Curia | 0.860 (± .005) | 0.856 (±.002) | 0.408 (± .022) | **0.748** (± .013) | **0.877** (± .011) |
| SuPReM | 0.777 (± .030) | 0.788 (± .025) | 0.341 (± .002) | 0.598 (± .004) | 0.598 (± .009) |
| CT-FM | **0.876** (± .011) | 0.847 (± .005) | 0.337 (± .015) | 0.589 (± .005) | 0.694 (± .009) |
| VISTA3D | 0.868 (± .021) | **0.866** (± .008) | 0.352 (± .005) | 0.603 (± .006) | 0.637 (± .011) |
| VoCo | 0.620 (± .004) | 0.632 (± .006) | 0.359 (± .004) | 0.610 (± .004) | 0.599 (± .050) |
| FMCiB | 0.776 (± .012) | 0.702 (± .054) | 0.347 (± .001) | 0.605 (± .015) | 0.587 (± .033) |
| TAP-B-3D | **0.876** (± .006) | 0.855 (± .007) | **0.420** (± .019) | 0.658 (± .022) | 0.798 (± .027) |

Table 5: Classification results of the TAP-CT ViT models: fine-tuning an Attention-Based Multiple Instance Learning (ABMIL) model on frozen `[CLS]` embeddings, when available, or on frozen patch embeddings otherwise. Each metric (AP, AUC) is averaged over three runs, with the corresponding standard deviation. **Best** results are bolded, while second-best results are underlined.

| Model | Patch Size | Image Size | LUNA16 (AUC) | LUNA25 (AUC) | RSNA2022 (AP) | RSNA2023 (AUC) | FDGPETCT (AUC) |
|---|---|---|---|---|---|---|---|
| TAP-S-2D | (16, 16) | (224, 224) | 0.830 (± .003) | 0.833 (± .005) | 0.390 (± .017) | 0.663 (± .015) | 0.789 (± .011) |
| TAP-B-2D | (16, 16) | (224, 224) | 0.854 (± .002) | **0.857** (± .003) | 0.385 (± .010) | 0.667 (± .006) | 0.757 (± .041) |
| TAP-S-2.5D | (1, 16, 16) | (6, 224, 224) | **0.886** (± .010) | 0.809 (± .003) | 0.439 (± .020) | **0.748** (± .007) | **0.889** (± .008) |
| TAP-B-2.5D | (1, 16, 16) | (6, 224, 224) | 0.815 (± .010) | 0.837 (± .003) | 0.429 (± .010) | 0.739 (± .010) | 0.827 (± .009) |
| TAP-S-3D | (4, 8, 8) | (12, 224, 224) | 0.868 (± .002) | 0.842 (± .005) | **0.440** (± .021) | 0.672 (± .016) | 0.805 (± .019) |
| TAP-B-3D | (4, 8, 8) | (12, 224, 224) | 0.876 (± .006) | 0.855 (± .007) | 0.420 (± .019) | 0.658 (± .022) | 0.798 (± .027) |
| **Ablation** | | | | | | | |
| TAP-B-3D (Patch Feat.) | (4, 8, 8) | (12, 224, 224) | 0.805 (±.029) | 0.855 (±.015) | 0.345 (±.003) | 0.592 (±.005) | 0.714 (±.033) |

This trend is less apparent when evaluating frozen features on volumes resized to ($z$, 512, 512), closer to native CT resolution. Base models still outperform smaller ones, but the 2D–3D gap narrows, likely because sliding-window inference remains limited by each model's native input size. Pretraining 3D models at 512×512 resolution would likely yield similar gains as observed at 224, but at substantially higher computational cost. A possible explanation for the stronger performance of the 2D and 2.5D models relative to the 3D models at 512-resolution is their larger patch extent in the $x$–$y$ plane rather than along z. Although all models have equal numbers of voxels per patch, resizing volumes to 512 solely affects the $x$–$y$ dimensions. As a result, the sliding-window evaluation may favor models whose patch structure allocates more capacity to these axes.

### 4.3. Classification

Table 4 summarizes the classification performance of TAP-B-3D compared to other pretrained CT foundation models. TAP-B-3D achieves the best results on two tasks (with one tie) and ranks second or third on the remaining three. Table 5 reports the classification

outcomes across the TAP-CT models, where TAP-S-2.5D attains the top performance on three tasks and demonstrates the overall strongest results, public models included. In contrast to the segmentation experiments, performance does not scale consistently with model size or dimensionality.

### 4.3.1. ANALYSIS

**Model comparison:** From Table 4, we observe that SuPReM's classification performance deteriorates substantially relative to its segmentation performance and compared to the other models. This can be attributed to its pretraining bias toward segmentation tasks, resulting in feature representations that are less suitable for classification. As with the segmentation tasks, VoCo's weaker performance on frozen feature classification can be attributed to the fact that the model is primarily designed to heavily fine-tune together with task-specific decoder. Although VISTA3D remains competitive on LUNA16 and LUNA25, its encoder, pretrained for pointwise segmentation, shows limited effectiveness on scan-level classification tasks (RSNA2022, RSNA2023, and FDGPETCT). In contrast, Curia and TAP-S-2.5D (Table 5) are able to extract meaningful signals from frozen features on these tasks. Given that Curia and TAP are the only models trained on private data and ViT/DINOv2 native, either could explain the performance difference, and further research is needed to draw conclusions. Moreover, the results in both Table 4 and Table 5 reveal a more fundamental challenge associated with volume-level classification in medical imaging. **Issues with volume-level classification in medical imaging:** Medical image classification often resembles finding a needle in a haystack: subtle, localized perturbations can decisively determine the outcome. For instance, diagnosing the presence of a lung nodule may depend on only a few voxels, and assessing whether the nodule is malignant poses an even greater challenge. Benchmarks such as LUNA16 tend to saturate quickly, as the classification task is typically confined to a predefined crop around the nodule when assessing malignancy. Furthermore, the clinical relevance of these benchmarks is limited, since they depend on prior nodule detection (and labeling) by a radiologist.

These challenges are clearly reflected in the results. On RSNA2022, most models struggle to surpass the expected average AP ($\approx 0.33$). For RSNA2023 and FDGPETCT, the ABMIL likewise fails to extract a strong signal from the majority of models. All models exhibit higher variance across runs in classification compared to segmentation tasks. Although TAP-S-2.5D and Curia generally perform well on volume-level tasks, the results among TAP-CT models show no consistent pattern. Increasing model size does not necessarily translate to improved outcomes, as the smaller variants frequently outperform their base counterparts. Furthermore, the difference between TAP-S/B-2D and TAP-S/B-3D remains marginal, suggesting that scaling dimensionality has limited influence in this context. Additionally, Curia's features appear capable of conveying global information, despite the model having access only to slice-level inputs. A potential limitation of the TAP-S/B-3D models may stem from compression constraints: while their large context benefits segmentation, it becomes challenging to encode global information into a single `[CLS]` token. These observations indicate that future research in CT foundation models should aim to extract global 3D representations while preserving robust and informative local features.

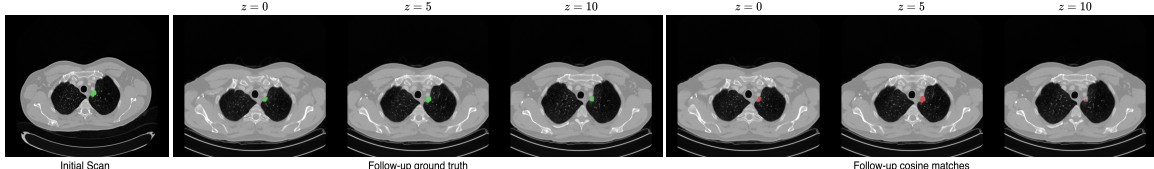

Figure 2: Cosine similarity matching between lesion embeddings in an initial and a follow-up scan of the same patient. The first set of slices are ground-truth lesion segmentations across ten slices ($z$); the second set shows the top-$k$ voxel matches between averaged lesion embeddings from the initial scan and all embeddings in the next.

### 4.4. Qualitative: Patch Retrieval

Tracking disease progression, such as changes in tumor volume over time, represents a potentially critical clinical application of pretrained models. This approach involves extracting a model's feature representation of a lesion from an initial scan and identifying the most closely related embeddings in subsequent scans. Ideally, features most relevant to the disease are consistently matched across follow-up scans. Figure 2 demonstrates that TAP-B-3D's features could be sufficiently descriptive to locate a lesion from an initial scan in a follow-up scan of the same patient. It indicates that fine-grained information is indeed present in the embeddings; however, a quantitative analysis would be needed to draw conclusions.

### 4.5. Why not use patch features for classification?

The frozen patch features of the TAP-CT models clearly encode rich semantic information, enabling effective segmentation with a simple decoder. In principle, the same information could be leveraged for classification using an attention-based mechanism such as ABMIL. However, it reflects the "needle-in-a-haystack" challenge: a single 300-slice CT scan processed by TAP-B-3D yields over 58,000 patches, making it difficult to isolate the few that are diagnostically relevant. Table 5 indeed demonstrates that scan-level classification deteriorates drastically when the ABMIL is applied to the frozen patch features, which indicates that global information retrieval is even more challenging from individual patches.

### 5. Conclusion

We introduced TAP-CT, a suite of 6 foundation models for CT imaging pretrained on 105K volumes through a novel 3D adaptation of the DINOv2 framework. This adaptation introduces a GPU-accelerated volumetric random resized crop and a 3D random masking strategy for DINO pretraining, alongside modifications of the patch embedding and positional encoding of a ViT. The models of TAP-CT achieve state-of-the-art performance on segmentation and competitive results on classification tasks. Therefore, self-supervised learning in CT should continue to focus on task-agnostic pretraining to develop well-rounded 3D vision encoders.

Further progress is needed to capture global information from volumetric medical data more effectively. Moreover, the requirement of multiple crops per sample in DINOv2 results in slow training and significant resource demands when scaling to volumetric data. These observations suggest an interesting direction for future work: developing less compute-intensive pretraining strategies that emphasize learning purely robust local features, followed by approaches to derive global representations from them effectively.

## Acknowledgments

We thank the Research High Performance Computing (RHPC) group at the Netherlands Cancer Institute, specifically Ameer Alkhier and Daniel Vis, for maintaining the GPU servers. Moreover, we like to thank the datadesk radiology, Artem Khmelinskii and Joost van Griethuysen, for downloading all CT scans from PACS.

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

| Model | GPU hours | Consumption (MWh) |
|---|---|---|
| **TAP-S-2D** | 96 | 0.067 |
| **TAP-B-2D** | 136 | 0.095 |
| **TAP-S-2.5D** | 384 | 0.269 |
| **TAP-B-2.5D** | 864 | 0.605 |
| **TAP-S-3D** | 1,152 | 0.806 |
| **TAP-B-3D** | 3,648 | 2.550 |
| **Total** | 6,280 | 4.396 |

Table 6: Overview of the total GPU hours required to train the TAP-CT models. One GPU hour corresponds to one hour of computation on a H100 SXM GPU with an approximate power draw of 700W.

## Appendix A. Pretraining Specifics

### A.1. Hyperparameters

A list of the hyperparameters used during the pretraining of TAP-B-3D is provided in Table 7. The global and local crop scales refer to the relative scale of the sampled height–width area, and a fixed number of slices along the $z$-axis are extracted by sampling around the corners of this area in the depth dimension. The pretraining hyperparameters for the ViT-S and ViT-B variants follow the same design, except that ViT-S uses a lower drop-path rate (0.1) to account for its smaller model size.

### A.2. GPU Hours

Table 6 reports the GPU hours and estimated energy consumption for pretraining the TAP-CT models. As expected, models with larger context sizes require substantially longer training times; for example, TAP-B-3D takes approximately 26 times longer to train than its 2D counterpart. Scaling in medical imaging remains a significant challenge. While training on entire volumes would be ideal, this is unlikely to be feasible in the near future. Therefore, developing methods that reduce computational requirements while maintaining high performance is essential.

## Appendix B. Dataset Specifics

In Figure 3, we visualize the distributions of scanner manufacturers, patient age groups, and patient sex. Because the dataset consists exclusively of individuals undergoing CT imaging for oncological assessment and tracking, the age distribution is naturally shifted toward older populations.

In Section 3, we note that foreground voxels are clipped and normalized using dataset-wide statistics. Specifically, we use: $\mu = -86.8086$, $\sigma = 322.6347$, $\text{clip}_{\min} = -1008.0$ and $\text{clip}_{\max} = 822.0$

## Appendix C. Downstream Task Specifics

### C.1. Batch size

Due to the variability in CT volume sizes, all downstream tasks are trained with a batch size of 1. To stabilize optimization, we employ 4 gradient accumulation steps for segmentation

Table 7: The hyperparameter configuration used for 3D DINOv2 pretraining of the model TAP-B-3D.

| TAP-B-3D DINOv2 Hyperparameters | | | |
|---|---|---|---|
| **Iterations** | 125,000 | **Scale Global Crops (Min, Max)** | (0.32, 1.0) |
| **DINO Loss Weight** | 1.0 | **Scale Local Crops (Min, Max)** | (0.05, 0.32) |
| **iBOT Loss Weight** | 1.0 | **Tied Head Weights** | No |
| **KoLeo Loss Weight** | 0.1 | **Head Prototypes** | 65536 |
| **Batch Size (Total)** | 2048 | **Head Hidden Dim** | 2048 |
| **Drop Path Rate** | 0.2 | **Head Layers** | 3 |
| **Layerscale** | 1e-5 | **Head Bottleneck Dim** | 256 |
| **Base Learning Rate** | 0.0035 | **Mask Probability** | 0.5 |
| **Weight Decay (Start, End)** | (0.04, 0.4) | **Mask Ratio (Min, Max)** | (0.1, 0.5) |
| **Teacher Momentum (Start, End)** | (0.992, 1.0) | **Centering** | Centering (No SK) |
| **Teacher Temperature (Start, End)** | (0.04, 0.07) | **ViT FeedForward Layer** | MLP |
| **Temperature Warmup Iterations** | 37,500 | **ViT Register Tokens** | 4 |
| **Gradient Clipping** | 3.0 | **Layerwise Decay** | 0.9 |

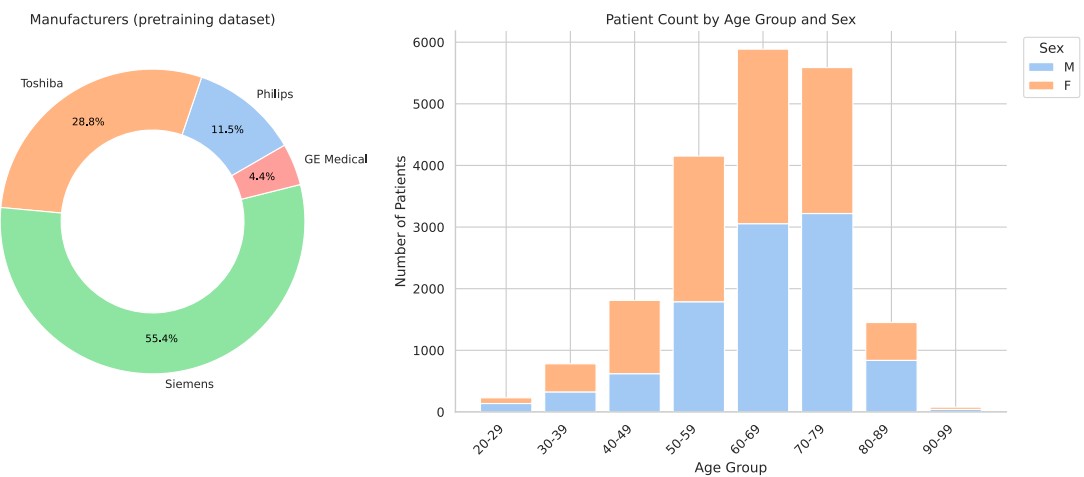

Figure 3: Distribution of pretraining data based on manufacturer, gender and age.

| TotalSegmentator Classes ||
| --- | --- |
| Background | Iliac Artery |
| Lungs | Iliac Vein |
| Kidneys | Humerus |
| Ribs | Scapula |
| Vertebrae | Clavicula |
| Spleen | Femur |
| Gallbladder | Hips |
| Liver | Spinal Cord |
| Stomach | Gluteus Maximus |
| Pancreas | Gluteus Medius |
| Adrenal Glands | Gluteus Minimus |
| Esophagus | Autochthon |
| Trachea | Iliopsoas |
| Thyroid Glands | Brain |
| Small Bowel | Skull |
| Duodenum | Sternum |
| Colon | Costal Cartilages |
| Urinary Bladder | Heart |
| Prostate | Aorta |
| Kidney Cysts | Pulmonary Vein |
| Sacrum | Brachiocephalic Trunk |
| Superior Vena Cava | Subclavian Artery |
| Inferior Vena Cava | Common Carotid Artery |
| Portal Vein and Splenic Vein | Brachiocephalic Vein |
| Atrial Appendage | |

Table 8: This table displays the individual classes of TOTALSEGMENTATOR used in the downstream task of this work. Originally there were 117 separate classes; these were merged into 49 classes, i.e. combining the different lobes of the lungs into a single lungs class.

tasks and 16 steps for classification tasks. For LUNA16 and LUNA25, the batch size is set to 64 because the input crops correspond to a fixed physical extent in millimeters around each lesion.

## C.2. Resampling

Each CT volume is resampled according to the preferred spacing of each publicly available model which can be found in Table 1. For our models, volumes are either resized to $(z, 224, 224)$ or $(z, 512, 512)$ in image space since we do not resample in world coordinates during pretraining.

## C.3. Merging TotalSegmentator

The TotalSegmentator v2 dataset comprises 117 anatomically distinct structures. Many of these labels correspond to fine-grained subdivisions of larger anatomical entities, such as individual bones that collectively form a unified structure. While such granularity is

valuable for detailed modeling, the distinction between specific vertebrae or ribs may be unnecessary for certain downstream tasks. In these cases, anatomically related subclasses can be merged into a single category without compromising the overall structural fidelity; these classes are found in Table 8.

### C.4. Omitting Classes RSNA2023

RSNA2023 includes bowel injury and extravasation as target abnormalities. However, the challenge organizers note that reliably identifying these findings is extremely difficult for radiologists without access to longitudinal follow-up imaging. Although the dataset provides coordinate annotations for a subset of scans, we omit these classes in our evaluation, as accurate volume-level classification is already highly challenging.

### C.5. Dataset splits

The dataset splits will be made available alongside the code upon acceptance.

### C.6. Qualitative visualization of segmentation performance

We show several abdominal CT slices with segmentation masks of the AMOS downstream task in Figure 4.

## Appendix D. Smaller local crops

Table 9: ViT-S-3D (Local Crops) performance across segmentation and classification tasks

| Model | Patch Size | Image Size | AMOS22 (DSC) | LiTS17 (DSC) | KiTS23 (DSC) | TotalSeg. (DSC) | MSD Pancreas (DSC) |
|---|---|---|---|---|---|---|---|
| | | | 0.482 ($\pm$ .001) | 0.504 ($\pm$ .001) | 0.381 ($\pm$ .006) | 0.427 ($\pm$ .003) | 0.278 ($\pm$ .002) |
| ViT-S-3D (Local Crops) | (1, 16, 16) | (12, 224, 224) | **LUNA16** (AUC) | **LUNA25** (AUC) | **RSNA2022** (AP) | **RSNA2023** (AUC) | **FDGPETCT** (AUC) |
| | | | 0.817 ($\pm$ .018) | 0.809 ($\pm$ .003) | 0.398 ($\pm$ .009) | 0.704 ($\pm$ .011) | 0.820 ($\pm$ .006) |

One of the primary factors influencing local-to-global correspondence in the DINOv2 framework is the relative dimensionality of local and global crops. In this work, the depths of the local and global crops are identical, which may bias the model toward the iBOT objective, as this configuration simplifies optimization of the DINO objective. Restricting local crops to a smaller number of slices therefore provides a straightforward way to encourage optimization toward the DINO objective. For this ablation, we train a 3D ViT-S with a patch size of (1, 16, 16), local crops of (6, 96, 96), and global crops of (12, 224, 224). The results for this model are reported in Table 9. Two observations can be made: (1) classification accuracy does not improve substantially relative to the other models, and (2) segmentation quality decreases significantly. The ablation model is outperformed by its 2D counterpart, despite having access to twelve times the context size. This reinforces the view that extracting 3D global representations from CT scans remains an open challenge and warrants further investigation in future research.

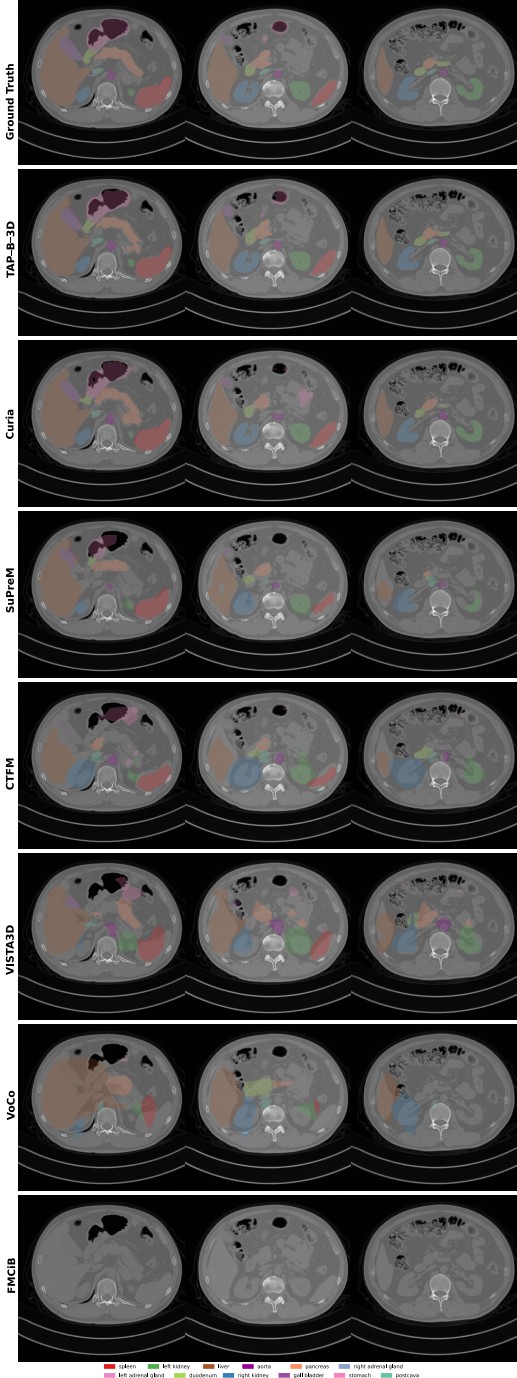

Figure 4: Segmentations across three abdominal slices from the AMOS22 validation sample (amos_286) for TAP-B-3D and other publicly available pretrained FMs. Each segmentation is produced using a linear convolutional layer fine-tuned on top of the frozen features of the respective pretrained encoder.

