# OpenReview forum: "TAP-CT: 3D Task-Agnostic Pretraining of Computed Tomography Foundation Models"
_MIDL.io/2026/Conference — MIDL 2026 Poster_

### Official Review · Reviewer_WWxv · 2025-12-24

**Confidence:** 4
**Preliminary Rating:** 4
**Final Rating:** 4

**Summary:**

The paper introduces TAP-CT, a framework of task-agnostic 3D foundation models pretrained on 105,000 CT volumes using an adapted DINOv2 framework. Experiments demonstrate that these 3D models yield robust frozen representations that achieve SOTA performance on segmentation benchmarks, outperforming existing baselines like Curia and SuPReM.

**Strengths:**

1. The pre-trained weights are open-sourced, facilitating community adoption and reproducibility.
2. All evaluations are implemented in EVA, ensuring fairness and consistency across experiments.
3. Extensive analysis of experimental results is provided to elucidate the observed phenomena.

**Weaknesses:**

1. The second paragraph of the introduction mentions two current applications of SSL. However, "methods that use the encoder only for specific tasks" appears to describe approaches with explicit training objectives, which should not be categorized as SSL.
2. The GPU-based 3D random resized crop does not appear to constitute a substantial technical innovation.
3. The comparison methods and related work section lacks several commonly used 3D CT foundation model baselines, including Merlin, CT-CLIP, Chest-OMDL, fVLM, RadZero3D, and DINOv3 ("Does DINOv3 Set a New Medical Vision Standard?").
4. In the local and global crops section, the specific novelty of the proposed cropping strategy remains unclear, as it appears fundamentally similar to conventional cropping approaches.
5. What is the source of the pretraining dataset? Is it proprietary or publicly available?
6. For all downstream tasks, would it be beneficial to include comparisons with baselines that do not leverage foundation models, thereby highlighting the value of pretraining?
5. Visualization for segmentation tasks appears limited, covering only organ segmentation scenarios.

**Detailed Comments:**

In Section 3.2, what specifically does "augmentations are replaced rather than omitted" refer to?

**Justification Of Final Rating:**

Thank you to the authors for addressing my concerns. The revised manuscript is clearer and more precise in its presentation. I believe the study has merit; therefore, I maintain my original score and recommend acceptance.

**Justification Of The Preliminary Rating:**

While the work exhibits certain limitations, it represents a thorough and solid contribution. The open-sourced weights provide valuable resources to the community. I recommend acceptance of this paper.

**Questions To Address In The Rebuttal:**

I suggest the authors consider the issues outlined in the weakness section, which could also be addressed in subsequent research.

---

> ### Author Response · Authors · 2026-01-25
> **Response to Reviewer WWxv**
>
> We appreciate the reviewer for their time and constructive feedback.
>
> **1. SSL definition**
>
> We thank the reviewer for this insight. We agree that approaches trained with explicit supervised objectives should not be categorized as SSL. The methods referenced in this paragraph do not necessarily rely on guided supervision during pretraining. For example, FMCiB (one of the pretrained encoders) is pretrained using a SimCLR objective, which remains self-supervised, but is applied to lesion-centered patches with the explicit goal of extracting imaging biomarkers.
>
> We have therefore placed greater emphasis in the introduction on distinguishing between the use of the terms foundation model and SSL. Moreover, we have added a sentence to explicitly clarify what we consider task-agnostic SSL in contrast to methods applied in other work.
>
> **2. GPU-based 3D Crop**
>
> We agree it is not a major algorithmic contribution, nor do we claim it as such. While technically straightforward, an efficient GPU-based implementation of RandomResizedCrop suitable for volumetric data is currently not available. This component is a necessary prerequisite for scaling DINOv2-style pretraining to 3D CT volumes, where CPU-based implementations would be prohibitively slow. We further discuss the design considerations between 2D and 3D cropping strategies in our response to the fourth point raised by the reviewer.
>
> **3. Extra comparison methods**
>
> Thank you for raising this point. While the listed methods are indeed commonly used in the literature, all except DINOv3 are vision-language models. In this work, we intentionally restrict our comparisons to vision-only encoders, as our goal is to benchmark the representational quality learned from imaging data alone, without the influence of language supervision. We agree that this scope was not made explicit in the original submission.
>
> We have now also clarified this in the revised manuscript within ‘Downstream Tasks’, Section 3.3. In addition, we now refer to DINOv2 and DINOv3 as natural-image pretrained vision-only baselines.
>
> **4. Novelty cropping approaches**
>
> We agree that cropping in 3D is fundamentally similar to cropping in 2D. The intent of the local and global crops section is therefore not to claim novelty in the cropping operation itself, but to motivate and clarify our choice of the specific cropping strategy used in this work. When translating random resized cropping from 2D to 3D, several design choices arise, and this section aims to discuss and justify those choices.
>
> In line with this clarification, we have revised the introduction to replace the term “reinterpreting” with “translating” when referring to the adaptation of DINOv2’s cropping strategy from 2D to 3D. We have also added an extra sentence to the ‘local and global crops’ section to emphasize that the section clarifies our motivation behind the 3D cropping strategy.
>
> **5. Dataset source**
>
> The pretraining dataset originates from the Netherlands Cancer Institute (NKI), a cancer research hospital located in Amsterdam, the Netherlands. Due to privacy regulations governing medical imaging data, the dataset is proprietary and not publicly available.
>
> However, to promote the reproducibility of our work, we have made all model weights available (https://huggingface.co/collections/fomofo/tap-ct) and will release all associated evaluation code.
>
> **6. Baselines**
>
> We agree that such comparisons would be beneficial. However, the objective of this work is not to achieve the best possible task-specific performance under full supervision, but rather to isolate and analyze the intrinsic transferability of the pretrained encoder. This setting allows us to assess the semantic structure learned by the FM itself, independent of extensive fine-tuning or decoder design. We view supervised pipelines such as nnU-Net as strong reference points in fully labeled regimes, and we do not claim that frozen TAP-CT models should match their performance, as this would need an investigation with a substantial decoder setup.
>
> We added clarification to the downstream task methodology (Section 3.3) and the future work discussion in the conclusion (Section 5.1) to more explicitly articulate the scope, motivation, and intended interpretation of the frozen-encoder evaluation.
>
> **7. Limited Visualizations**
>
> We have included more segmentation maps of the TotalSegmentator validation set, alongside the abdominal CT scan from AMOS22. Section C.6 now contains separate visualizations for both slices from the thorax and the head.
>
> **8. Augmentations**
>
> The augmentations in the original DINOv2 are domain-specific, i.e. color jittering. However, for CT (single-channel), color jittering will not have any effect on the data. Therefore we’ve opted for the replacement of natural image augmentations with transforms more suited to CT (e.g. gamma adjustment). We have removed the ‘rather than omitted’ from the manuscript to mitigate confusion for future readers.

---

### Official Review · Reviewer_m2Py · 2025-12-29

**Confidence:** 5
**Preliminary Rating:** 2
**Final Rating:** 3

**Summary:**

The paper introduces TAP-CT - task agnostic DINOv2 style pretraining for 3D CT volumes. The pretraining happens on private in-house dataset of 105k 3D volumes. The authors evaluate the quality of learned representations in frozen setting attaching and finetuning only lightweight task-specific heads for various downstream tasks (segmentation and classfication). The paper also includes qualitative analyses of patch-level retrieval and lesion tracking to illustrate the semantic structure captured by the pretrained model. Overall, the work aims to provide insights into what 3D foundation models learn from large-scale CT data and how these representations behave when adapted minimally to different tasks.

**Strengths:**

1. The paper addresses the growing need for large-scale, task-agnostic 3D pretraining methods in medical imaging, an area gaining significant attention with the rise of foundation models. Adapting DINOv2 to volumetric CT data is well motivated, and the scale of pretraining (105k CT volumes) is relatively large compared to existing 3D SSL efforts.

2. The choice to probe the pretrained model using frozen features and a single lightweight head is conceptually meaningful. This setup provides insight into the quality and "knowledge" of the learned features, independent of task-specific fine-tuning capacity, and aligns with the way many general-purpose vision foundation models are evaluated.

3. The authors test TAP-CT on diverse segmentation and classification benchmarks, including organ-level and lesion-level tasks (TotalSegmentator, LiTS, KiTS, LNs), which helps highlight the strengths and limitations of the learned representations across task types.

4. Patch-retrieval and lesion-tracking visualizations help illustrate what kinds of volumetric correspondences the model captures. These qualitative results support the idea that DINO-style training yields semantically meaningful feature spaces even without fine-tuning.

5. The last but absolutely not least - releasing the pretrained model weights and simple snippet code is valuable for the community and can serve as a strong baseline for future 3D foundation model research.

**Weaknesses:**

1. The frozen encoder results across all downstream tasks are substantially lower than the fully supervised nnU-Net baselines. While the authors state that full fine-tuning is out of scope, the size of the performance gap raises important questions about the practical utility of using TAP-CT in a frozen form. In a realistic clinical setting, segmentation or classification systems operating at the accuracy levels observed in the frozen experiments would rarely be considered sufficient, especially when well-established architectures (e.g., nnU-Net) can achieve considerably higher performance with relatively modest fine-tuning cost.

Given this discrepancy, it becomes unclear when or why strictly frozen evaluation should be the preferred operating point for a volumetric foundation model. If the goal is to demonstrate the intrinsic quality of the learned representations, then linear-probe evaluation is informative; however, if the goal is to argue for practical clinical applicability, frozen results alone may be misleading. The paper would benefit from a clearer articulation of the intended use case: is TAP-CT meant to be a representational probe (similar to linear probes in natural-image foundation models), or is it meant to support task adaptation under realistic constraints?

Maybe the direction the paper seems to be moving toward (“one encoder for many tasks”) naturally suggests an alternative evaluation setting such as one-shot or few-shot learning. These scenarios would more directly highlight the value of strong pretrained volumetric features: the encoder remains mostly frozen, but a very small amount of task-specific data is used to adapt the head. This would allow the authors to maintain the “minimal adaptation” philosophy while producing results that are more meaningful and more aligned with real-world medical workflows, where limited annotated data is often available but fully training a model from scratch is still costly.

Hence, a clarification or a small additional experiment in a one-shot/few-shot regime would therefore strengthen the framing of the paper and help position TAP-CT more clearly within the landscape of medical-image foundation models.

2. Limited comparison of pretraining strategies. The paper exclusively evaluates DINO-based pretraining, but does not compare against MAE- (for local representations) or Contrastive- (for global representations) style reconstruction methods. Especially, since many downstream tasks rely on local structures, the comparison to MAE style pretraining (or an justification for its non inclusion) feels important.

3. Compute cost justification is unclear. TAP-B-3D requires extremely high GPU hours to train, yet task-specific fine-tuning with nnU-Net can achieve strong performance with just ~12 GPU hours. It remains unclear how the substantial pretraining cost is justified across tasks, especially when frozen performance lags behind supervised baselines.

**Detailed Comments:**

1. Clarify the motivation for using only frozen encoder evaluation.
The paper consistently evaluates TAP-CT by freezing the pretrained encoder and training only a shallow task-specific head. While this probing approach is informative for analyzing representation quality, the resulting performance is far lower than fully supervised baselines (e.g., nnU-Net). A brief explanation of why the frozen setting is emphasized would help readers understand the intended use case. If the goal is to highlight the intrinsic semantic structure of the representations, or to support extremely low-resource adaptation, this should be stated more explicitly. As the paper aspires toward “one model for all tasks,” you may also consider whether one-shot or few-shot learning would be a more aligned evaluation protocol, since it preserves minimal adaptation but yields more practically meaningful performance levels.

2. Add a short discussion comparing DINO-style and MAE-style self-supervision.
TAP-CT adopts DINOv2-style training, which is known to encourage strong global semantic consistency. By contrast, MAE pretraining encourages richer local spatial representations, which often benefit volumetric segmentation tasks. Recent work (e.g., Tassilo Wald et al., Revisiting MAE Pre-training for 3D Medical Image Segmentation) has investigated 3D MAEs for medical contexts. Even if running a full MAE baseline is not feasible, it would be helpful to add a short discussion explaining why DINOv2 was selected over MAE and what trade-offs the authors expect between the two.

3. Provide a clearer explanation of the reported pretraining compute cost.
The TAP-B-3D variant requires thousands of GPU hours to train, whereas task-specific fine-tuning with nnU-Net often requires on the order of 10-15 GPU hours per dataset. It would be helpful to understand the reasoning of this gap by explaining how the pretraining cost is reducing the finetuning costs across many downstream tasks, or by discussing how smaller TAP variants behave in terms of cost-performance trade-offs. Even a brief paragraph would make the resource implications of TAP-CT easier to interpret.

4. Consider noting how TAP-CT compares to nnU-Net.
nnU-Net Revisited (Isensee et al., MICCAI 2024) offers improved and standardized baselines for CT segmentation. Even if you do not add new experiments, a short mention in the discussion could help position TAP-CT more accurately within the current state of supervised 3D medical segmentation.

5. A compact dataset summary table (volumes, modalities, label types) would make the experimental setup more accessible.

**Justification Of Final Rating:**

Addressing the review comments substantially improved the manuscript. The added large-scale evaluations, foundation-model comparisons, and expanded OOD analysis strengthen the paper and better position the proposed approach within the current landscape of medical foundation models.That said, I still do not see a fully convincing justification for the practical use of pretrained CT models under low-compute or frozen-adaptation settings, as performance remains substantially below fully fine-tuned baselines. While the representational analysis and few-shot experiments are informative, the gap to supervised pipelines suggests that current foundation models are not yet at a stage where linear probing alone constitutes a competitive alternative for most clinical tasks. For this reason, while I appreciate the improvements and view the work as promising, I slightly revise my rating to borderline, reflecting both the strengthened empirical contribution and the remaining uncertainty regarding near-term practical impact.

**Justification Of The Preliminary Rating:**

While the paper presents an interesting and ambitious attempt to build a task-agnostic 3D CT foundation model, the current evaluation does not clearly demonstrate a practical use case in the settings represented by the tested downstream datasets. The frozen-encoder paradigm provides insight into the learned representations, but without a clearer articulation of when such frozen performance would be operationally meaningful or why it should be preferred over inexpensive task-specific fine-tuning, the applicability of TAP-CT remains unclear. As a result, it is difficult to see how the proposed model would actually be deployed or relied upon in realistic clinical or research workflows. I view the core ideas as promising, and the paper could become impactful with a stronger framing of its target usage scenario, but in its current form the practical relevance is not sufficiently established.

**Questions To Address In The Rebuttal:**

1. Can you better justify the emphasis on frozen-encoder evaluation? In what practical scenarios is frozen TAP-CT expected to be preferred over task-specific fine-tuning, given the current performance gap?
2. Can you clarify how the high pretraining compute cost compares to the cumulative cost of fine-tuning models across many tasks?
3. Is it feasible to include a small-scale MAE or CNN-based frozen baseline for comparison?

---

> ### Author Response · Authors · 2026-01-25
> **Response to Reviewer m2Py**
>
> We thank the reviewer for their thoughtful input and feedback.
>
> **1. Can you better justify the emphasis on frozen-encoder evaluation? In what practical scenarios is frozen TAP-CT expected to be preferred over task-specific fine-tuning, given the current performance gap?**
>
> We thank the reviewer for raising this concern. The emphasis on frozen-encoder evaluation arises from the goal of assessing foundation models purely based on the quality of their learned representations, independent of task-specific fine-tuning. We acknowledge that the original manuscript did not sufficiently justify the use of frozen-encoder evaluation. To address this, we have revised the downstream task methodology (Section 3.3) to provide a more thorough explanation of this evaluation choice.
>
> Frozen TAP-CT is currently expected to be preferred in zero-labeled data to low-labeled data regimes. To support this claim, we have added a few-shot organ classification experiment (Sections 3.3.2 and 4.4), which relies solely on similarity between prototype features of different organ classes in the AMOS22 dataset.
>
> **2. Can you clarify how the high pretraining compute cost compares to the cumulative cost of fine-tuning models across many tasks?**
>
> We agree with the reviewer that the cost of pretraining is a concern that should be addressed. The compute cost of foundation models should ideally be viewed as a one-time, amortized investment. Fine-tuning models across many downstream tasks can be effective when large amounts of annotated data are available; however, medical imaging is well known to suffer from data scarcity due to the expertise and time required for annotation. In addition, deploying multiple independently trained models introduces significant practical challenges, as each model must be individually validated and assessed for robustness and safety in a clinical setting.
>
> At the same time, current medical imaging foundation models remain far from direct clinical applicability. We have added extra discussion on the pretraining compute cost versus current applications of supervised learning in medical image in the conclusion.
>
> **3. Is it feasible to include a small-scale MAE or CNN-based frozen baseline for comparison?**
>
> Adding an MAE-based baseline would indeed provide an interesting comparison. However, setting up and training such a baseline was not feasible within the review period.
>
> We have added an in-depth discussion in the introduction on existing SSL-methods based on their properties (i.e. generative through inpainting, next-token prediction and contrastive, such as view-invariance learning) to substantiate our choice for the DINOv2 framework as task-agnostic SSL pretraining method.
>
> **4. Consider noting how TAP-CT compares to nnU-Net. nnU-Net Revisited (Isensee et al., MICCAI 2024) offers improved and standardized baselines for CT segmentation [...] supervised 3D medical segmentation.**
>
> We agree and have updated the manuscript accordingly in ‘Conclusion Section 5.1’. We now explicitly position nnU-Net (and nnU-Net Revisited) as the current practical standard for fully supervised CT segmentation and a strong upper bound in settings where dense annotations are available. We note that while TAP-CT achieves state-of-the-art performance among frozen, task-agnostic foundation models, it is not intended to replace supervised nnU-Net pipelines for day-to-day clinical segmentation tasks at present. Instead, TAP-CT is designed to provide transferable representations that can support label-efficient learning, rapid adaptation across tasks, and standardized benchmarking of CT foundation models.
>
> We additionally note in Section 3.3.1 that our downstream segmentation benchmarks (AMOS, KiTS, LiTS, TotalSegmentator, MSD Pancreas) overlap with those recommended in nnU-Net Revisited, and we now cite this work to better contextualize our results within the current 3D medical segmentation landscape.
>
> **5. A compact dataset summary table (volumes, modalities, label types) would make the experimental setup more accessible.**
>
> We have now added a compact dataset summary in ‘Appendix C: Downstream Task Specifics’ under ‘C.5: Dataset Summary and Dataset Splits’ and refer to it from the main methodology Section 3.3.

---

> > ### Comment · Reviewer_m2Py · 2026-01-27
> > **Follow-up on Point 4**
> >
> > One remaining suggestion would be to also include fully supervised baseline(s) directly in Table 2 (e.g., best performing standard CNN training from scratch). While the conceptual discussion is helpful, presenting these numbers side-by-side with frozen representations + LP results would make the performance gap immediately visible and allow readers to better "appreciate" the trade-off between linear probing on pretrained representations and task-specific supervised training.
> > Currently, the results suggest that foundation models for CT imaging are still not at a stage where linear probing alone approaches the performance of supervised baselines. Given this gap, it would strengthen the paper to more explicitly articulate the concrete benefits or use cases of operating in a frozen or low-adaptation regime (for now 5 segmentation datasets in table 2 do not support that claim).

---

> > > ### Author Response · Authors · 2026-01-29
> > > **Response to Reviewer m2Py**
> > >
> > > Thank you for the response. We would like to clarify that the primary goal of Table 2 is not to argue that frozen or low-adaptation regimes are already competitive replacements for supervised CNN training in routine segmentation workflows. Rather, Table 2 is intentionally framed as a representation benchmark, aimed at isolating and evaluating the intrinsic quality and generality of the learned representations produced by the foundation models currently available within the community.
> > >
> > > We do agree with the reviewer that it is important to contextualize the gap between supervised learning and linear probing of pretrained models. Therefore, we have updated the manuscript with a supervised 3D nnU-Net ResEnc (M) baseline for all segmentation datasets and have reported them in Table 2 as suggested. Additionally, Section 4.1 highlights that the performance of frozen pretrained weights with linear probes is considerably lower than that of fully optimized encoder-decoder setups like nnU-Net.

---

### Official Review · Reviewer_mnxg · 2026-01-09

**Confidence:** 5
**Preliminary Rating:** 4
**Final Rating:** 4

**Summary:**

This paper presents an adaptation of the DINOv2 self-supervised framework to 3D CT volumes with the goal of learning a general-purpose foundation model for medical imaging. The main modifications concern the handling of patch and class tokens, the cropping strategy for 3D inputs, and the use of the pretrained encoder for both segmentation and classification downstream tasks. The work includes a thorough evaluation protocol, with comparisons against recent self-supervised pretraining baselines, multiple ablations to support the design choices and interpretations of the results

**Strengths:**

1) The pretraining setup leverages a substantial dataset (approximately 105k CT scans), which is likely to improve representation quality and robustness
2) Adaptation of the state-of-the-art DINOv2 model to the 3D CT images.
3) The paper includes extensive experiments, ablation studies, and interpretability-oriented analyses, leading to clear and useful conclusions about the effects of the proposed design choices

**Weaknesses:**

1) Algorithmic novelty is limited to the adaptation of DINOv2 to 3D images. However, design choices are thoroughly justified and supported.
2) Pretraining is performed on a private dataset, which may limit reproducibility and complicate direct comparison with other state-of-the-art methods.
3) Because the approach is closely tied to the DINOv2 paradigm, the learned representations appear stronger for segmentation than for classification. In particular, the underperformance on some classification settings (as indicated in Table 4) weakens the claim that the model is broadly task-agnostic without additional analysis or mitigation strategies.

**Detailed Comments:**

Some tables and figures are placed far from the sections where they are first referenced, which makes the manuscript harder to follow. If possible, please rearrange the layout so that key tables/figures appear closer to the relevant text to improve readability.

**Justification Of Final Rating:**

The authors have successfully addressed the specific questions raised in my initial review.
Regarding the paper's significance, I maintain my original assessment: while the algorithmic novelty is limited to adapting the DINOv2 framework, the application, training methodology and thorough evaluation represent a clear and practically important contribution.
Therefore, I recommend weak acceptance.

**Justification Of The Preliminary Rating:**

I recommend weak accept because the paper makes a clear and practically important contribution: it adapts a strong self-supervised framework (DINOv2) to 3D CT and supports the proposed design choices with extensive experiments, ablations, and comparisons to relevant self-supervised baselines. Although the algorithmic novelty and contribution are limited to adapting DINOv2, the scale of pretraining and the thorough evaluation suggest the resulting representations are valuable.

**Questions To Address In The Rebuttal:**

1) Given that DINOv3 is now available, do the authors anticipate that adopting DINOv3-style improvements could further enhance performance in future work
2) The manuscript mentions window-based inference in some settings. Please specify the configurations used (e.g., window size, overlap/stride, aggregation strategy)
3) Elaborating on the classification underperformance (Tables 4 and 5): Can the authors better explain the main factors behind the slight underperformance on some classification tasks? Is this related to biases of DINO-style objectives that may favor dense/local representations, beneficial for segmentation?

---

> ### Author Response · Authors · 2026-01-25
> **Response to Reviewer mnxg**
>
> We thank the reviewer for the response and the constructive feedback.
>
> **1. Given that DINOv3 is now available, do the authors anticipate that adopting DINOv3-style improvements could further enhance performance in future work?**
>
> We agree that the post-training strategies introduced in DINOv3 are promising for future research on CT foundation models. Specifically, pretraining at low resolution for most iterations, followed by a short high-resolution pretraining phase of only a few thousand iterations, seems promising. Given that high spatial resolution is a fundamental challenge in CT modeling due to increased dimensionality, such a strategy is especially appealing and can be explored in future work.
>
> We have now added a sentence in the ‘The Scaling of TAP-CT’ section describing this anticipation.
>
> **2. The manuscript mentions window-based inference in some settings. Please specify the configurations used (e.g., window size, overlap/stride, aggregation strategy)**
>
> This should have been described indeed, we thank the reviewer for highlighting it. We have now specified this in the Methods section:
>
> Window-based inference: we set the window size to the native input size of each encoder, and use 0.75x overlap per dimension, equivalent to 0.25x striding window. Overlapping embeddings are merged via Gaussian averaging.
>
> **3. Elaborating on the classification underperformance (Tables 4 and 5): Can the authors better explain the main factors behind the slight underperformance on some classification tasks? Is this related to biases of DINO-style objectives that may favor dense/local representations, beneficial for segmentation?**
>
> We expect this behavior to arise from a combination of factors. First, classification in medical imaging is inherently challenging, as labels are often determined by subtle, localized patterns within very high-resolution data. Second, the DINO objective encourages global information to be aggregated into the CLS token. As input resolution increases, the number of patch tokens grows substantially, requiring the model to compress increasingly rich global information into the fixed-capacity CLS token. In combination with the first point, this compression can be detrimental, as the task requires sensitivity to small, fine-grained signals.
>
> This perspective could explain why the 2.5D models outperform both the 2D and 3D variants on classification tasks: they retain more contextual information than 2D models, while requiring less global compression than full 3D models. We have added this extra hypothesis to the “Classification Analysis Section 4.3.1”.
>
> **4. Some tables and figures are placed far from the sections where they are first referenced, which makes the manuscript harder to follow. If possible, please rearrange the layout so that key tables/figures appear closer to the relevant text to improve readability.**
>
> We thank the reviewer for the suggestion to improve readability. We have moved the tables to their respective sections and stacked the results tables for both segmentation and classification to facilitate easier comparison between different TAP-CT variants and publicly available pretrained models.

---

> > ### Author Response · Authors · 2026-01-29
> > **Response to Reviewer mnxg**
> >
> > We thank the reviewer again for their careful evaluation and feedback. We hope that our responses and revisions address the raised questions satisfactorily, and we remain available for any further clarification during the remainder of the discussion period.

---

### Author Rebuttal · Authors · 2026-01-25

**Rebuttal:**

We thank the reviewers for their constructive feedback. We appreciate the suggestions regarding 1) positioning and scope (frozen evaluation; comparison set), 2) classification underperformance, 3) compute cost interpretation, and 4) clarifications around window-based inference, augmentations, and dataset provenance. We have focused our attention on improving those, with more details in the specific responses. We have also attached our revised manuscript; the changes are highlighted in green.

Edited during discussion period: On request of reviewer m2Py, we have updated Table 2 with a supervised baseline and interpretation of the results in Section 4.1 accordingly.

**Supporting Material:**

/attachment/10409cc2849620018d0b9fda70cb2aa39e47e1da.pdf

---

### Meta-Review · Area_Chair_j1Sh · 2026-02-10

**Recommendation:** Accept (Poster)
**Confidence:** 4

**Metareview:**

The paper presents a task-agnostic 3D CT foundation model by adapting DINOv2 to volumetric data. Two reviewers are positive, while one remains borderline. Overall, the contribution is technically sound and clearly relevant to MIDL. I recommend acceptance (poster); however, the framing, presentation clarity, and depth of analysis do not yet support an oral presentation.

---

### Decision · Program_Chairs · 2026-02-13

Accept (Poster)